# Identification of Genetic Polymorphisms of PI, PIII, and Exon 53 in the Acetyl-CoA Carboxylase-α (ACACα) Gene and Their Association with Milk Composition Traits of Najdi Sheep

**DOI:** 10.3390/ani13081317

**Published:** 2023-04-12

**Authors:** Abdulkareem M. Matar, Abdulrahman S. Alharthi, Moez Ayadi, Maged A. Al-Garadi, Riyadh S. Aljummah

**Affiliations:** Department of Animal Production, College of Food and Agriculture Sciences, King Saud University, P.O. Box 2460, Riyadh 11451, Saudi Arabia

**Keywords:** milk composition, fatty acid profile, acetyl-CoA carboxylase-α gene, SNP, Najdi breed

## Abstract

**Simple Summary:**

The acetyl-CoA carboxylase gene (ACACα) plays a role in facilitating the delivery of fatty acid precursors to the mammary glands during lactation. The purpose of this study was to identify single-nucleotide polymorphisms (SNPs) for promoter I (PI), promoter III (PIII), and Exon 53 in the *ACACα* gene to determine whether SNPs are associated with milk composition (MC) and the fatty acid profile (FA) in Najdi sheep milk. An analysis and alignment of the PI, PIII, and Exon 53 sequences for the *ACACα* gene identified twenty SNPs. The homozygous ewes with PI, PIII, and Exon 53 SNPs were associated (*p* < 0.05) with essential fatty acids (EFAs) (α-linolenic acid (ALA, n3), linoleic acid (LA, n6), and conjugated fatty acid (CLA)) in their milk fat. These results suggest that *ACACα* genes are involved in PUFA synthesis, which modifies the compositional properties of Najdi sheep milk, resulting in healthier dairy products.

**Abstract:**

Recently, increasing attention has been paid to sheep milk products, which are high in saturated fatty acids (SFA), and the extent of their impact on human health. This study aimed to identify SNPs for PI, PIII, and Exon 53 in the *ACACα* gene and their association with the MC and FA profiles in Najdi sheep milk. A total of 76 multiparous Najdi ewes were used, and they were maintained using the same feeding system. Milk and blood samples were collected during the first lactation. A genetic polymorphism analysis identified 20 SNPs: 4 SNPs on PI, 6 SNPs on PIII, and 10 SNPs on Exon 53. In PI, the SNP g.4412G > A was associated (*p* < 0.05) with palmitic acid (C16:0), palmitoleic acid (16:1 n-7) and linoleic acid (LA), while SNP g.4485C > G was associated with CLA and vaccenic acid (VA) (*p* < 0.05). Furthermore, in PIII, two SNPs (g.1168A > G and g.1331G > T) were associated with milk protein (*p* < 0.05), while the SNP g.6860G > C in Exon 53 was associated with milk fat (*p* < 0.05). SNPs in the Najdi breed have been shown to be strongly related to milk fat and EFA contents. This could support a genetic selection program and the control of milk traits in the Najdi breed of high-quality dairy sheep.

## 1. Introduction

The acetyl-CoA carboxylase gene is one of the important genes affecting the MC and FA profile in milk fat [1]. This paves the way for the future optimization of sheep breeding programs and requires extensive genetic research. Recently, increasing attention has been paid to the fat content and fatty acid (FA) profile of sheep milk and the magnitude of its impact on human health, as the quality of milk and dairy products is critical to meeting consumer needs. Mozaffarian, et al. [2] showed that palmitoleic acid (C16:1, PA) and CLA are metabolic health biomarkers and that FA and phospholipid levels are associated with insulin resistance, atherogenic dyslipidemia, and the development of type 2 diabetes. Furthermore, milk composition and the FA profile are crucial components in determining the nutritional value of milk.

The synthesis of fatty acids in milk underlies the role of several enzymes such as *ACAC*, *SCD*, *FAS*, and di-glyceride acyltransferase [3]. There are two types of *ACAC* enzymes in mammals: *ACACα* and *ACACβ*. *ACACα* is an enzyme involved in lipid metabolism and storage, and ACACβ acts as a mitochondrial enzyme for the synthesis of malonyl-CoA, altering fatty acid oxidation by suppressing fatty acid transport to the mitochondria [4]. *ACAC* plays a crucial role in the catalysis of malonyl-CoA, the main substrate for promoting fatty acid synthesis and elongation as well as limiting fatty acid oxidation [5]. *ACAC* is produced in high concentrations in lipogenic tissues such as the liver, adipose tissue, and mammary glands [6].

Sheep were sequenced for the *ACACα* gene (cDNA 6.6 kb) located on chromosome 11 (OAR11), which has 54 Exons, and 12 SNPs showed higher gene frequency in dairy cultivars [7]. Compared to other species, the *ACACα* gene in sheep has many variations. The majority of studies have focused on the promoter regions of the *ACACα* gene, with promoter I (PI) and promoter II (PII) identified in mammals [8] and promoter III (PIII) found in sheep [9] and goats [10].

The PI transcript is mainly present in adipose tissue, while the PII domain transcript is found in almost every tissue [9], and the PIII transcript is mainly found in lactating mammary tissues [6]. Exons are the parts of genes that code for amino acids, and Exon 53 in the Valle del Belice sheep breed is the most polymorphic, with 12 SNPs, indicating that the *ACACα* gene is extremely variable in sheep [11]. In addition, expression of the *ACACα* gene in mammary glands increases 15–28-fold during lactation [9,12].

The Najdi sheep is a desert-adapted native breed of fat-tailed sheep that is recognized for its hardiness and adaptability. Najdi sheep are mainly found in the central and eastern regions of Saudi Arabia. Under improved feeding and management practices, the Najdi breed performs satisfactorily and has high potential for milk production with intensive production [13]. The aim of this study was to identify genetic polymorphisms in the promoters (PI and PIII) and Exon 53 of the *ACACα* gene in the Najdi sheep population and to identify SNPs associated with milk quality traits.

## 2. Materials and Methods

### 2.1. Animals, Management, and Experimental Design

In this study, 76 ewes were randomly selected from a private Al- Khaldiyah farm in Riyadh. All ewes were multiparous and were reared on the same farm to avoid variations in management conditions. They were fed a diet consisting of 30% alfalfa hay and 70% concentrate without additives, as presented in Table 1. Feed was provided ad libitum, and the animals had free access to fresh clean water. All animals were tested for mastitis using a California mastitis test (CMT) to ensure the animals were healthy. The lactating Najdi ewes that were studied had body weights of 61.71 ± 0.96 kg and milk production of 0.749 ± 0.347 L/d. In addition, the body condition score (BCS) of the ewes was measured at the beginning of the study (3.37 ± 0.55).

### 2.2. Collection and Analysis of Milk Samples

All ewes were in the first stage of lactation (<30 days in milk; DIM) and were milked at in the morning (08:00 h). Samples of milk (50 mL) were collected from the whole milk after the morning milking. The composition of the milk (fat, protein, and lactose) was analyzed 24 h after collection using a Milko-Scan FT6000 (Foss, Hillerød, Denmark), and the milk was then stored at −20 °C until the analysis of the FA profiles. The extracted lipids were obtained from 30 mL milk samples according to Luna, et al. [14], and FA methyl esters (FAMEs) were prepared according to [15,16]. The relative proportions of each FA profile were calculated based on the ratio of the peak area of one FA to the total peak area of all FAs in the fat sample. In total, 30 fatty acids were determined as follows: C6:0, C8:0, C10:0, C12:0, C14:0, C15:0 iso, C15:0 antiso, C15:0, C16:0, C17:0, C16:1 (n-9), C16:1 (n-7), C17:0 antiso, C17:0, C17:1 cis10, C18:0, C18:1 (n-9), C18:1 (n-7) cis-11, C18:1 cis-13, C18:1 cis-14, C18:2 (n-6), C18:2 (n-6), C20:0, C18:3 (n-3), C18:2 cis-9, trans-11, C21:0, C22:0, C20:4 (n-6), C22:4 (n-6), and C22:5 (n-3).

### 2.3. Blood Sampling and DNA Extraction

Blood samples (10 mL) were collected from all animals using jugular vein puncture and EDTA vacuum tubes and were stored at −4 °C until DNA extraction. Genomic DNA was extracted from the whole blood of all ewes using a GFX genomic blood DNA 27-9603-01 100-purification kit (GE Healthcare, Chicago, IL, USA) according to the manufacturer’s instructions. The integrity and purity of the extracted DNA samples were assessed using 0.8% gel electrophoresis and a Nano-Drop 2000/C spectrophotometer (Thermo Scientific), respectively. The obtained O.D. 260/280 ratios ranged from 1.8 to 2.2, indicating high-quality DNA.

### 2.4. Amplification of the ACACα Gene via Polymerase Chain Reaction (PCR)

Specific primers targeting PI [17], PIII [18], and Exon 53 [11] were used to amplify the *ACACα* gene (Table 2). For the final volume of the PCR, a 20 µL reaction contained 10 µL of Taq Hot Start Green Master Mix (Promega), 1 µL of each primer (*ACACα* forward and reverse: 0.5 µM), and 4 µL of DNA, and nuclease-free water was added to a total volume 20 µL. The PCR conditions were as follows: initial denaturation at 95 °C for 5 min; 35 cycles of amplification in three stages, including denaturation at 95 °C for 30 s, hybridization at 58–61 °C for 45 s, and extension at 72 °C for 40 s; and a final extension incubation at 72 °C for 10 min. The *ACACα* gene PCR product was checked using 2% agarose gel electrophoresis containing ethidium bromide and was recorded using a gel documentation system (Syngene Synoptic Ltd., Cambridge, UK).

### 2.5. Sequencing and Genotype Analysis

In total, 76 samples were sequenced in both the forward and reverse directions. The primers (forward and reverse) for sequencing were the same as those used for the PCR amplification. The *ACACα* gene promoter I, promoter III, and Exon 53 PCR products were purified and sequenced by the Macrogen sequencing service (Macrogen Inc., Seoul, Republic of Korea). The sequences were edited, aligned using the Geneious 5.5.9 version Biomatters Ltd. software, and compared to identify polymorphic sites. The obtained sequences were compared with their respective reference sequences (GenBank accession numbers; PI: AJ292285.1, PIII: AJ292286, and Exon 53: ENSOARG00000000829).

### 2.6. Analyses of the Associations and Effects of Alleles

The allele frequency and genetic equilibrium of the studied population were estimated using the Hardy–Weinberg equilibrium and were tested using the chi-square test for the 76 ewes online at https://wpcalc.com/en/equilibrium-hardy-weinberg/ (accessed on 17 June 2021). The effect analyses for the SNPs at PI, PIII, and Exon 53 of the *ACACα* genes for milk composition (fat, protein, and lactose) and the FA profiles of milk fat were performed in Najdi sheep using the Proc Mixed procedure in SAS software (SAS-P.4 Institute Inc., Cary, NC, USA) [19].

y_ijklm_ = µ + Age_i_ + BI_j_ + SNP_kl_ + Animal_m_ + e_ijklm_,

where y_ijklm_ is the dependent variable; µ is the overall mean; Age_i_ is a fixed effect of an animal’s age at calving; BI_j_ is a fixed effect of type of birth with two levels (single and twins); SNP_jl_ is a fixed effect of *ACACα* gene genotype; Animal_m_ is a random effect of the m_th_ individual ewe nested within the *ACACα* genotype; and e_ijklm_ is a residual error. Significance was declared when *p* < 0.05.

### 2.7. Linkage Disequilibrium (LD) Estimation

Haploview 4.2 software (Broad Institute of MIT and Harvard, Cambridge, MA, USA) was used to estimate the extent of linkage disequilibrium (LD). The associations of genotypes and haplotypes with the MC and FA profile were tested using g-PLINK (version 1.9; Broad Institute of MIT and Harvard).

In this study, we demonstrated the effective sample size and the statistical power required to achieve 80% statistical power, as calculated using G*Power 3.1.7 software, for the genotypes of the *ACACα* gene under various assumptions related to milk quality traits according to Hong and Park [20] and Gauderman [21].

## 3. Results

### 3.1. Sequence Analysis of the ACACα Gene Coding Region

The PCR products for the three DNA fragments (PI: 377 bp, PIII: 526 bp, and Exon 53: 499 bp) were separated using agarose gel electrophoresis within 60 min, as shown in Figure 1. The genotypes and frequencies of the least frequently analyzed alleles are given in Table 3 and Table 4. No significant variation was found for any SNPs on PI, PIII, and Exon 53 in the Najdi sheep *ACACα* gene using the Hardy–Weinberg frequencies equilibrium. The detection of SNPs from sequencing read-outs and a multiple sequence alignment of the Najdi sheep *ACACα* gene (PI, PIII, and Exon 53) revealed four SNPs in PI, including g.4412G > A, MT512519; g.4441T > A, MT512520; g.4485C > G, MT512521; and g.4507T > C, MT512522 (Table 3 and Appendix A), and two nucleotide insertions of A and G at positions 4579 and 4602, respectively, with the GenBank accession number MT512523.

The genomic regions of PIII (499 bp) in the *ACACα* gene, from 956 bp to 1454 bp, were sequenced. The sequencing revealed six SNPs (g.1007T > G, MT512524; g.1014C > T, MT512525; g.1168A > G, MT512526; g.1331G > T, MT512527; g.1339C > G, MT512528; and g.1431C > T, MT512529) and a nucleotide insertion of T at position 1184 in all individuals (according to GenBank accession number MT512530) (Appendix A). Two of these SNPs (g.1007T > G and g.1014C > T) and the nucleotide insertion of T at position 1184 in PIII in the ACACα gene of the Najdi breed were described for the first time (Table 3).

A sequence analysis of 526 bp at Exon 53 in the *ACACα* gene revealed ten SNPs with the GenBank accession numbers shown in Table 4. Five of these were first recorded in the Najdi breed (g.6627G > C, g.6668C > T, g.6908G > A, g.7029G > T, and g.7031C > T) and have not been reported in any other breed in the sequence references (LT627649 and LT627657) (Appendix A).

In addition, an 18 bp sequence of AC/GGTGAGTATGCGGCCC was inserted at position 6852. Using the ExPASy translate tool to translate the sequence of Exon 53 showed that two of these SNPs are predicted mutations that cause amino acid substitutions: g.6860G > C (serine/TCA to stop codon/TGA) and g.7031C > T (replaces the amino acid threonine/ACG to methionine/ATG, according to GenBank accession number QST04603.1). The allele frequencies and genotypes for all SNPs detected in Exon 53 (identified and linked to accession numbers) are summarized in Table 4.

### 3.2. Association of Genotype of ACACα Gene with Milk Traits

The results presented in Table 3 and Table 4 summarize the SNPs that showed significant (*p* < 0.05) associations with the MC and FA profiles. An association analysis showed that the SNP g.4412G > A of PI in the *ACACα* gene was significantly (*p* < 0.05) associated with palmitic acid (C16:0), palmitoleic acid (16:1 n-7), and LA (C18:2 n-3) (Table 5), while SNP g.4441T > A was significantly (*p* < 0.05) associated with behenic acid (C22:0), and SNP g.4485C > G was significantly (*p* < 0.05) associated with CLA (C18:3) and VA (C18:1). In addition, SNP g.4507C > T showed a significant (*p* < 0.05) association with arachidic acid (C20:0). It is worth noting that the homozygous ewes with AA at SNP 4412G > A and CC at SNP g.4485C > T had a significant effect on LA, VA, and CLA compared to heterozygous ewes.

The association analysis of SNPs, revealed that SNP g.1007T > G at PIII in the *ACACα* gene was significantly (*p* < 0.05) associated with lauric acid (C12:0) and CLA, while SNP g.1014C > T was significantly (*p* < 0.05) associated with CLA. In the same context, heterozygous ewes with CT and GT alleles for the SNPs 1007T > G and 1014C > T had a significant (*p* < 0.05) effect on CLA compared to homozygous ewes (Table 5). Furthermore, the SNPs g.1168A > G and g.1331G > T were significantly (*p* < 0.05) associated with the protein content of milk and palmitoleic acid (16:1n-7). Whereas SNP g.1431C > T was significantly (*p* < 0.05) associated with CLA and arachidonic acid (C20:4). Notably, milk from heterozygous ewes with AG and GT alleles at SNP g.1168A > G had high protein contents (4.58 ± 0.21 and 4.56 ± 0.20) compared to homozygous ewes.

Overall, 9 of the 10 SNPs identified at Exon 53 of the *ACACα* gene in Najdi sheep were significantly associated with at least one milk trait (Table 6). The six SNPs at Exon 53, namely g.6627G > C, g.6855C > T, g.6894T > C, g.6898T > C, g.6989C > G, and g.7031C > T, were significantly (*p* < 0.05) associated with LA (C18:2 n-6) and arachidic acid (C20:0), as shown in Table 6. In addition, the SNPs g.6894T > C and g.6898T > C showed significant (*p* < 0.05) associations with ALA (C18:3 n-3).

Interestingly, the SNP g.6860G > C at Exon 53 was significantly (*p* < 0.05) associated with the milk fat content of Najdi ewes. The heterozygous ewes with the GC and CC alleles showed lower milk fat levels (3.04 ± 0.40 and 3.19 ± 0.58, respectively) compared to the homozygous ewes with the GG allele at SNP g.6860G > C (4.16 ± 0.22). It is worth noting that the homozygous ewes whose SNP alleles at Exon 53 were GG, CC, or TT were superior in their EFA contents, such as LA (C18:2 n-6) and ALA (C18:3 n-3).

The LD ranges between the SNPs at PI, PIII, and Exon 53 of the *ACACα* gene, which were estimated using Haploview 4.2, were divided into three blocks comprising 6, 7, and 10 haplotypes, respectively. The pairwise D’ measurements showed that the SNPs were strongly linked within the blocks (Block 1: D’= 1; Block 2: D’= 0.95 to 1; and Block 3: D’ = 1) (Figure 2).

The frequencies of the six haplotypes in block 1 were as follows: H1B1 (TCAGCC: 41%), H2B1 (TCGTGC: 39%), H3B1 (GTGTCT: 8%), H4B1 (GCGTCT: 5%), H5B1 (TTGTCT: 3%), and H6B1 (TCGTGT: 1%). The frequencies of the seven haplotypes in block 2 were as follows: H1B2 (GTCT: 39%), H2B2 (GTGT: 31%), H3B2 (ATGC: 12%), H4B2 (ATCT: 7%), H5B2 (AAGC: 4%), H6B2 (ATCC: 4%), and H7B2 (AACC: 3%). In addition, the frequencies of the 10 haplotypes in block 3 were as follows: H1B3 (GGTCCTGC: 42%), B3H2 (GCTCCTTC: 19%), H3B3 (CGCTGCGT: 15%), H4B3 (CGCTGCGT: 1%), H5B3 (CGTCGCGT: 2%), H6B3 (CGTCCTGC: 7%), H7B3 (GGTCCTTC: 1%), H8B3 (CGCCGCGT: 1%), H9B3 (CGCCGCGC: 1%), and H10B3 (CGCTGCTT: 3%), as shown in Figure 3.

An association analysis between the haplotype blocks with MC and FA profilers showed significant (*p* < 0.05) associations with milk fat, C6:0, C8:0, ALA, LA, C16:1 cis7, and C20:0 (Table 7). Haplotype block1 was significantly (*p* < 0.05) associated with C16:1 cis7 and C22:0, while haplotype block 2 was significantly (*p* < 0.05) associated with C6:0, C8:0, and ALA. In addition, haplotype block 3 was significantly (*p* < 0.05) associated with the milk fat percentage and LA (Table 7). The results in Table 7 summarize the haplotypes in the different blocks that showed significant associations with all milk compositions and fatty acid profiles.

## 4. Discussion

### 4.1. Sequence Analysis of the ACACα Gene Coding Region

Several previous studies suggested that *ACACα* is a promising candidate gene for milk composition traits in sheep, and this follow-up study revealed significant genetic associations of *ACACα* with milk fat and PUFAs [22,23,24]. Until recently, there were few reports on the *ACACα* gene in sheep and little data on the SNPs associated with milk composition traits in sheep [7,25].

The SNPs g.4485C > G and g.4507T > C in the PI of the *ACACα* gene in Najdi sheep reported in the present study are identical to a previous finding in three Italian sheep breeds, Gentile di Puglia, Sarda, and Altamurana, [22] and in British sheep [6,9]. On the other hand, in other species such as goats, eight SNPs have been identified in the PI region [26], while in cattle 28 SNPs have been identified in the PI of the *ACACα* gene [27].

Three SNPs were identified in the PIII of the *ACACα* gene in Italian sheep [22] and Munjal sheep [25], namely g.1330G > T, g.1338C > G, and g.1430T > C, but these SNPs were in different positions compared to the results of the current study. In addition, similarities with the results of this study were reported in the East Friesian breed [18]. SNPs in PIII have also been reported in Holstein cattle (1956C > G; 1597C > T; and 1256C > T) [27] and goats [10]. These results indicate that the different SNP positions in the DNA promoter gene sequence in dairy animals can affect the binding transcription factor (TF) and that miRNA targets can alter gene expression [28].

In the current study, most of the SNPs identified in Exon 53 and the sequence insertion (18 bp) at position 6852 of the *ACACα* gene in Najdi sheep were similar to those found in Vall del Belice sheep [11]. Interestingly, five SNPs, g.6627G > C, g.6668C > T, g.6908G > A, g.7029G > T, and g.7031C > T, were reported for the first time in Najdi sheep. The results of the present study were a culmination of the emergence of a new sequence that can radically alter the resulting proteins and cause a mutation similar to a stop codon, as mentioned by Gerlando et al. [11]. The sequence of the *ACACα* gene and the SNPs identified in sheep [11,24] overlapped with the current study, which identified SNPs in PI and PII of the *ACACα* gene expressed in Najdi sheep.

The translated sequences of Exon 53 showed differences in translated protein in Najdi sheep compared to Valle del Belice sheep [11]. Previous studies have reported that the activity of the encoded enzyme could be related to SNPs [11,24], which may explain the results of this study due to the presence of two SNPs on Exon 53, causing amino acid substitution, as mentioned previously [11].

### 4.2. Association of Genotype of ACACα Gene with Milk Traits

In the current study, the SNPs identified in PI, PIII, and Exon 53 of the *ACACα* gene showed strong associations with the milk fat and FA contents in Najdi sheep milk. Several studies have been performed to identify SNPs in the ACAC genes of sheep [10,22] and goats [29]. Furthermore, few studies have been conducted to determine the relationship between SNPs and the MC or FA profile of sheep milk [7,23,30]. SNPs g.1168A > G and g.1331G> in PIII were found to have significant associations with the milk protein content, while SNP g.6860G > C in Exon 53 had a strong association with the milk fat content in Najdi sheep. Similar to the results of the current study, it was observed that the SNP 1330G > T in PIII of the *ACACα* gene in Italian sheep had a significant impact on the milk fat content [23]. Furthermore, SNP 1430C < T in PIII of the *ACACα* gene in Munjal sheep had a positive effect on milk fat and protein content [25].

A study in Holstein–Friesian and Japanese black cattle reported that three SNPs in the PIII of the *ACACα* gene have significant associations with long-chain FAs (C14:0, C16:0, and C18:0) [3]. In addition, a report on cattle found that SNPs in the PI of the *ACACα* gene were significantly associated with the FAs (C14:0, C14:1, C16:1, and C18:1 trans) of the longissimus dorsi muscle [26]. Furthermore, Kęsek, et al. [31] found that *ACACα* polymorphism had marked effects on the C13:0, C14:1, C16:1, and CLA levels in Holstein–Friesian cows. 

In the current study, the majority of SNPs in Exon 53 had significant effects on ALA (C18:3-n3), LA (C18:2-n6), and CLA, while the LD for block 3 haplotypes was strongly associated with LA. This confirms the previous statement that most studies on the effect of *ACACα* on de novo FA synthesis in the mammary glands found that the *ACACα* gene was associated with PUFAs that are not synthesized by milk-producing cells [32]. Polyunsaturated fatty acids are essential fatty acids that are not endogenously synthesized and must be obtained from the diet. Nevertheless, there is evidence that the levels of C18:2 and C18:3 FAs in milk is regulated not only by dietary factors but also by genetic factors since the heritability of the PUFA levels in milk is low to moderate (C18:2, h^2^ = 0.11 to 0.27; C18:3, h^2^ = 0.09) [33]. Kong, et al. [34] mentioned that the PUFA metabolism and lipid deposition in longissimus dorsi muscle fat can be strongly influenced by *ACACα*. In contrast, the FASN, ACAC, and SCD polymorphisms are key genes affecting PUFA composition and are promising for improving FA composition [34,35]. Most of the previously reported candidate genes show strong effects on milk traits, but the mechanisms by which these genes control milk quantity or composition have yet to be investigated [36].

In a study by Kęsek-Woźniak et al. [32], it was found that homozygous cows with GG at SNP g.1488C > G in the *ACACα* gene had significant differences in C10:0, C12:0, C14:0, and C15:0 compared to the GC and CC alleles. SNPs in the gene’s promoter can alter transcription factor binding, which affects promoter activity and can be considered a functional mutation [3]. This indicates that SNPs in PI and PIII in the *ACACα* gene may play a crucial role in regulating transcript expression in mammary epithelial cells, thereby affecting FA metabolism and explaining the role of the *ACACα* gene in milk FA biosynthesis [3,6,7]. This result, found in Brown Swiss cows with SNP A > G in the *ACACα* gene, was negatively associated with de novo synthesized short- and medium-chain FAs: C8:0, C10:0, and C12:0 [37]. Dettori, et al. [38] reported that the SNP of Exon 9 of the *ACACα* gene showed that the ewes with the CC genotype had a lower somatic cell count and a significant correlation with the clotting time.

This demonstrated that the *ACACα* enzyme is important in the de novo synthesis of FAs in the mammary gland. The results of the current study confirm the effect of the *ACACα* enzyme on UFA biosynthesis, which is mainly controlled by the animal’s diet. Therefore, the results of this study are expected to show that the ACAC gene is associated with lipid metabolism properties. Finally, the results of this study indicate that the SNP found in PI is associated with palmitic acid (C16:0), which could be attributed to the reported biological function of the *ACACα* gene [7] since the ACAC gene is a lipid synthesis restriction enzyme.

## 5. Conclusions

A total of 20 SNPs of the Najdi breed *ACACα* gene were identified via an analysis and an alignment of sequences. The SNP g.6860G > C (MT649199) in Exon 53 of the *ACACα* gene significantly influenced the milk fat content. Furthermore, SNPs in PI, PIII, and Exon 53 were significantly associated with EFAs, particularly the LA-n3, ALA-n6, and CLA in the milk fat of Najdi sheep. The identified association between the Najdi breed genotypes and phenotypes could play a crucial role in altering the milk composition and FA profile of milk fat and could form the basis of a genetic selection program to produce healthy milk. Further research is needed to examine the association of the *ACACα* gene with the fatty acid profile in dairy sheep using a large sample.

## Figures and Tables

**Figure 1 animals-13-01317-f001:**
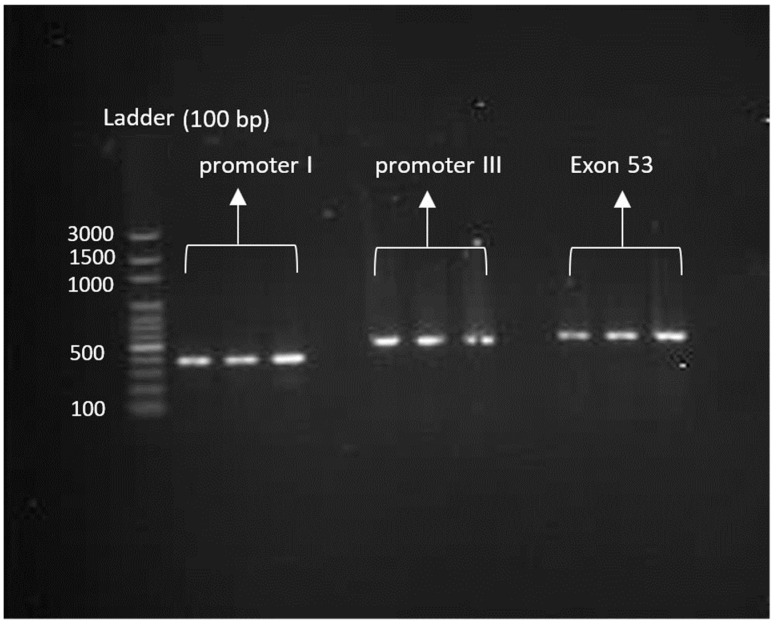
PCR products of *ACACα* gene of Najdi sheep in 2% gel electrophoresis agarose (PI: promoter I; PIII: promoter III, and E53: Exon 53).

**Figure 2 animals-13-01317-f002:**
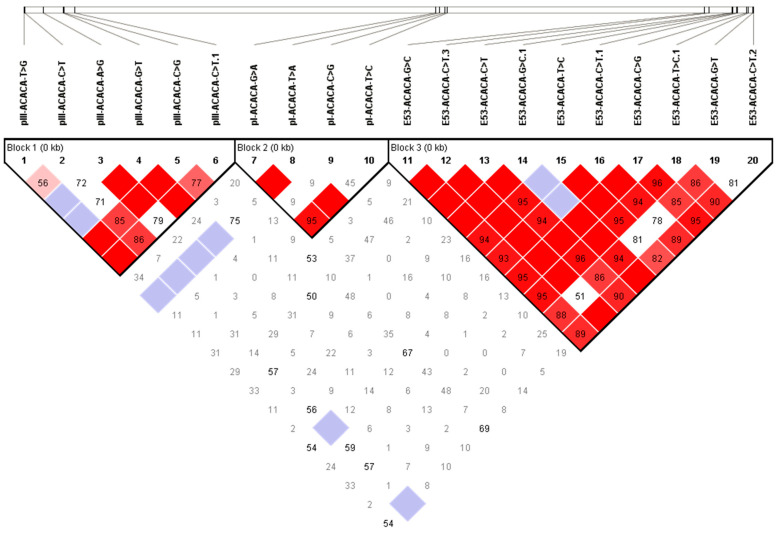
Estimation of the linkage disequilibrium between SNPs in the *ACACα* gene. The blocks indicate haplotype blocks, and the text above the horizontal numbers indicates the SNP names. The values in the boxes are pairwise SNP correlations (r^2^). The SNPs not in bold indicate that SNPs do not form a haplotype block with other SNPs.

**Figure 3 animals-13-01317-f003:**
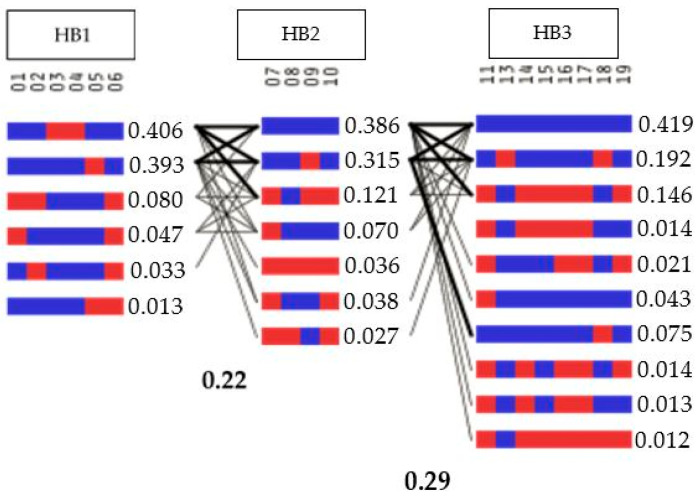
Three blocks of 21 haplotypes (HB) with different forms were identified in the *ACACα* gene of Najdi sheep: HB1 (6 haplotypes), HB2 (7 haplotypes), and HB3 (10 haplotypes).

**Table 1 animals-13-01317-t001:** The chemical composition and fatty acid profile of the traditional forage (alfalfa hay) and concentrate based on dry matter.

Nutrition	30% Alfalfa Hay	70% Concentrate
Chemical composition (%)		
Dry matter %	27.41	62.97
Crude protein %	5.19	9.11
ME, Mcal/kg	0.82	2.01
NDF %	13.29	26.11
ADF %	10.97	16.87
Ash %	3.138	8.44
Fat %	0.002	1.71
Fatty acid composition (%)		
C6:0	1.14	--
C8:0	3.12	0.12
C12:0	0.48	--
C14:0	1.83	0.12
C16:0	22.66	15.04
C16:1 cis 9	1.29	0.18
C17:0	0.76	0.12
C18:0	6.26	2.29
C18:1 trans 11	--	1.29
C18:1 cis 9	10.20	23.70
C18:2 cis 9, 12	17.42	51.43
C20:0	3.72	0.39
C18:3 cis 9, 12, 15	25.32	4.93
C22:0	3.92	0.29
C20:4 cis7, 10, 13, 16	1.87	0.10
SFA	43.89	18.37
UFA	56.11	81.63

SFA, saturated fatty acid; UFA, unsaturated fatty acid; ME, metabolism energy; NDF, neutral detergent fiber; ADF, acid detergent fiber.

**Table 2 animals-13-01317-t002:** PCR primers used to sequence promoter I, promoter III, and Exon 53 in the Najdi sheep *ACACα* gene.

Target	Sequences	Annealing Temperature (°C)	PCR (bp)
Promoter I	F: GTGGCAAACGTTGTCTTTCT R: CGTATGGGCTTCACTGACTG	60°	377
Promoter III	F: GGGCCCCTTTACGTTTCTGT R: AGCTTCTGCCTTAGCTGCA	61°	526
Exon 53	F: CCAGTTATCAGCAGAGGCGG R: GTGGGACTCAGTTTCCCGTC	60°	499

**Table 3 animals-13-01317-t003:** Genotype and allele frequencies in single-nucleotide polymorphisms (SNPs) at promoter I and promoter III of the *ACACα* gene in Najdi sheep.

SNPs	Acc. No.	Allele Frequencies		Hardy–Weinberg Frequencies			Chi-Squared (χ^2^)
PI region							
4412G > A	MT512519	G = 0.69	A = 0.31	GG (36) *E* = 36.96	AG (34) *E* = 32.07	AA (6) *E* = 6.96	0.27
4441T > A	MT512520	T = 0.92	A = 0.08	TT (64) *E* = 64.47	AT (12) *E* = 11.05	AA (0) *E* = 0.47	0.55
4485C > G	MT512521	C = 0.34	G = 0.66	CC (8) *E* = 8.89	CG (36) *E* = 34.21	GG (32) *E* = 32.89	0.21
4507T > C	MT512522	T = 0.79	C = 0.21	TT (42) *E* = 40.97	TC (20) *E* = 22.06	CC (14) *E* = 2.97	0.57
PIII region							
1007T > G	MT512524	T = 0.86	G = 0.14	TT (57) *E* = 56.45	GT (15) *E* = 18.09	GG (3) *E* = 1.45	2.15
1014C > T	MT512525	C = 0.86	T = 0.14	CC (59) *E* = 56.45	CT (11) *E* = 11.95	TT (5) *E* = 1.45	11.69
1168A > G	MT512526	A = 0.43	G = 0.57	AA (11) *E* = 13.90	AG (42) *E* = 37.20	GG (22) *E* = 24.89	1.84
1331G > T	MT512527	G = 0.41	T = 0.59	GG (10) *E* = 12.64	TG (41) *E* = 36.71	TT (24) *E* = 26.64	1.57
1339C > G	MT512528	C = 0.58	G = 0.42	CC (27) *E* = 25.47	GC (33) *E* = 37.05	GG (15) *E* = 13.47	0.51
1431C > T	MT512529	C = 0.81	T = 0.19	CC (50) *E* = 49.76	TC (21) *E* = 23.46	TT (4) *E* = 2.76	0.84

SNP: Single-nucleotide polymorphisms; Acc. No.: GenBank accession number.

**Table 4 animals-13-01317-t004:** Genotype and allele frequencies in single-nucleotide polymorphisms (SNPs) at Exon 53 of the *ACACα* gene in Najdi sheep.

SNPs	Acc. No.	Allele Frequencies		Hardy–Weinberg Frequencies			Chi-Squared (χ^2^)
Exon 53 region							
6627G > C	MT649197	G = 0.69	C = 0.31	GG (38) *E* = 36.26	CG, (29) *E* = 32.46	CC (9) *E* = 7.26	0.86
6668C < T	MT649197	C = 0.79	T = 0.21	CC (46) *E* = 45.03	TC (25) *E* = 26.94	TT (5) *E* = 4.03	0.39
6855T > C	MT649198	T = 0.74	C = 0.26	TT (45) *E* = 41.26	TC (22) *E* = 29.47	CC (9) *E* = 5.26	4.88
6860G > C	MT649199	G = 0.76	C = 0.24	GG (49) *E* = 44.26	CG (18) *E* = 27.47	CC (9) *E* = 4.26	9.04
6894T > C	MT649200	T = 0.76	C = 0.24	TT (48) *E* = 44.26	TC (20) *E* = 27.47	CC (8) *E* = 4.26	5.62
6898C > T	MT649201	C = 0.80	T = 0.20	CC (53) *E* = 48.16	TC (15) *E* = 24.67	TT (8) *E* = 3.16	11.68
6977C > G	MT649202	C = 0.73	G = 0.27	CC (44) *E* = 40.52	CG (23) *E* = 29.94	GG (9) *E* = 5.53	4.08
6989T > C	MT649203	T = 0.74	C = 0.26	TT (44) *E* = 41.26	TC (24) *E* = 29.47	CC (8) *E* = 5.26	2.62
7029G > T	MT649204	G = 0.68	T = 0.32	GG (38) *E* = 34.89	GT (27) *E* = 33.20	TT (11) *E* = 7.89	2.65
7031C > T	MT649205	C = 0.77	T = 0.23	CC (49) *E* = 45.02	TC (19) *E* = 26.94	TT (8) *E* = 4.02	6.60

SNP: Single-nucleotide polymorphisms; Acc. No.; GenBank accession number.

**Table 5 animals-13-01317-t005:** The associations of the SNPs of PI and PIII in the *ACACα* gene with the milk composition (%) and fatty acid profiles (g/100 g) in Najdi sheep milk.

Milk Traits	Genotypes (Mean ± SE)			*p* Value
*Promoter I*				
Locus SNP: 4412G > A	GG	AG	AA	
C16:0	27.33 ± 0.56 ^a^	26.68 ± 0.57 ^b^	24.35 ± 1.21 ^c^	0.04
C16:1 cis7	0.29 ± 0.01 ^c^	0.30 ± 0.01 ^b^	0.34 ± 0.02 ^a^	0.03
C18:2 cis9, 12 (LA)	3.91 ± 0.19 ^c^	4.01 ± 0.19 ^b^	4.80 ± 0.32 ^a^	0.01
Locus SNP: 4441T > A	TT	TA	--	
C22:0	0.13 ± 0.02 ^b^	0.21 ± 0.03 ^a^	--	0.01
Locus SNP: 4485C > G	CC	CG	GG	
C18:1cis11 (VA)	0.53 ± 0.02 ^a^	0.49 ± 0.02 ^b^	0.48 ± 0.02 ^c^	0.05
C18:2 (CLA)	0.82 ± 0.03 ^a^	0.71 ± 0.03 ^c^	0.76 ± 0.03 ^b^	0.03
Locus SNP: 4507C > T	CC	TC	TT	
C20:0	0.23 ± 0.02 ^c^	0.32 ± 0.01 ^a^	0.30 ±0.01^b^	0.01
*Promoter III*				
Locus SNP: 1007T > G	GG	GT	TT	
C12:0	4.22 ± 0.62 ^a^	2.83 ± 0.33 ^c^	3.19 ± 0.28 ^b^	0.04
CLA	0.49 ± 0.10 ^b^	0.77 ± 0.04 ^a^	0.76 ± 0.02 ^ab^	0.03
Locus SNP: 1014C > T	CC	CT	TT	
CLA	0.75 ± 0.02 ^b^	0.82 ± 0.04 ^a^	0.57 ± 0.07 ^c^	0.02
Locus SNP: 1168A > G	AA	AG	GG	
Protein	4.08 ± 0.25 ^b^	4.58 ± 0.21 ^a^	4.51 ± 00.21 ^ab^	0.04
C16:1 cis7	0.28 ± 0.01 ^c^	0.31 ± 0.01 ^a^	0.29 ± 0.01 ^b^	0.005
Locus SNP: 1331G > T	GG	GT	TT	
Protein	4.07 ± 0.25 ^b^	4.56 ± 0.20 ^a^	4.54 ± 0.20 ^a^	0.05
C16:1 cis7	0.27 ± 0.01 ^c^	0.31 ± 0.01 ^a^	0.30 ± 0.01 ^b^	0.004
Locus SNP:1431C > T	CC	TC	TT	
CLA	0.76 ± 0.03 ^ab^	0.77 ± 0.03 ^a^	0.56 ± 0.08 ^b^	0.05
C20:4	0.31 ± 0.02 ^b^	0.35 ± 0.02 ^a^	0.27 ± 0.04 ^c^	0.03

VA: vaccenic acid; LA: linolenic acid; CLA: conjugated fatty acid; SE: standard error. Values with different superscripts within a row are significantly different at *p* ≤ 0.05.

**Table 6 animals-13-01317-t006:** The associations of the SNPs at Exon 53 in the *ACACα* gene with the milk composition (%) and fatty acid profiles (g/100 g) in Najdi sheep milk.

Traits	Genotypes (Mean ± SE)			*p* Value
Locus SNP: 6627G > C	GG	CG	CC	
C18:2 cis9, 12 (LA)	4.11 ± 0.16 ^a^	3.89 ± 0.17 ^b^	3.49 ± 0.27 ^c^	0.05
C20:0	0.31 ± 0.01 ^b^	0.28 ± 0.01 ^c^	0.35 ± 0.02 ^a^	0.001
C22:0	0.14 ± 0.01 ^b^	0.12 ± 0.01 ^c^	0.15 ± 0.01 ^a^	0.04
C22:5	0.18 ± 0.01 ^a^	0.15 ± 0.01 ^c^	0.16 ± 0.02 ^b^	0.01
Locus SNP: 6855C > T	CC	TC	TT	
C16:1 cis7	0.32 ± 0.01 ^a^	0.28 ± 0.01 ^c^	0.30 ± 0.01 ^b^	0.03
C18:2 cis9, 12 (LA)	3.50 ± 0.26 ^c^	3.79 ± 0.18 ^b^	4.11 ± 0.15 ^a^	0.02
C20:0	0.35 ± 0.02 ^a^	0.28 ± 0.01 ^c^	0.30 ± 0.01 ^b^	0.01
Locus SNP: 6860G > C	GG	GC	CC	
Fat	4.16 ± 0.22 ^a^	3.04 ± 0.40 ^c^	3.19 ± 0.58 ^b^	0.03
C20:0	0.30 ± 0.01 ^b^	0.29 ± 0.01 ^c^	0.35 ± 0.02 ^a^	0.04
C22:0	0.13 ± 0.01 ^b^	0.12 ± 0.01 ^c^	0.18 ± 0.01 ^a^	0.001
Locus SNP: 6894T > C	TT	CT	CC	
C18:2 cis9, 12 (LA)	4.09 ± 0.14 ^a^	3.82 ± 0.18 ^b^	3.25 ± 0.28 ^c^	0.01
C20:0	0.31 ± 0.01 ^b^	0.27 ± 0.01 ^c^	0.37 ± 0.02 ^a^	0.001
C18:3 (ALA)	0.84 ± 0.06 ^a^	0.75 ± 0.07 ^b^	0.68 ± 0.09 ^c^	0.05
C21:0	0.08 ± 0.01 ^b^	0.07 ± 0.01 ^c^	0.09 ± 0.01 ^a^	0.03
C22:0	0.14 ± 0.01 ^b^	0.11 ± 0.01 ^c^	0.15 ± 0.02 ^a^	0.01
Locus SNP: 6898T > C	TT	TC	CC	
C18:2 cis9, 12 (LA)	3.28 ± 0.28 ^c^	3.74 ± 0.21 ^b^	4.08 ± 0.14 ^a^	0.01
C20:0	0.38 ± 0.02 ^a^	0.28 ± 0.01 ^c^	0.30 ± 0.01 ^b^	0.006
C18:3 (ALA)	0.68 ± 0.09 ^c^	0.73 ± 0.08 ^b^	0.84 ± 0.06 ^a^	0.04
Locus SNP: 6977C > G	CC	CG	GG	
C16:1 cis7	0.30 ± 0.01 ^b^	0.28 ± 0.01 ^c^	0.32 ± 0.01 ^a^	0.03
C18:2 cis9, 12 (LA)	4.13 ± 0.15 ^a^	3.78 ± 0.17 ^b^	3.50 ± 0.26 ^c^	0.01
C20:0	0.30 ± 0.01 ^b^	0.28 ± 0.01 ^c^	0.35 ± 0.02 ^a^	0.01
Locus SNP: 6989C > T	CC	CT	TT	
C18:2 cis9, 12 (LA)	3.26 ± 0.28 ^c^	3.95 ± 0.17 ^b^	4.06 ± 0.14 ^a^	0.02
C20:0	0.37 ± 0.02 ^a^	0.28 ± 0.01 ^c^	0.30 ± 0.01 ^b^	0.004
Locus SNP: 7029G > T	GG	GT	TT	
C16:0	29.3 ± 0.50 ^a^	28.1 ± 0.58 ^b^	26.6 ± 0.91 ^c^	0.03
C20:0	0.30 ± 0.01 ^b^	0.29 ± 0.01 ^c^	0.34 ± 0.02 ^a^	0.03
C21:0	0.08 ± 0.01 ^b^	0.07 ± 0.01 ^c^	0.10 ± 0.01 ^a^	0.05
C22:0	0.13 ± 0.01 ^b^	0.12 ± 0.01 ^c^	0.17 ± 0.01 ^a^	0.001
Locus SNP: 7031C > T	CC	TC	TT	
C18:2 cis9, 12 (LA)	4.08 ± 0.15 ^a^	3.86 ± 0.19 ^b^	3.28 ± 0.28 ^c^	0.01
C20:0	0.29 ± 0.01 ^c^	0.30 ± 0.01 ^b^	0.37 ± 0.02 ^a^	0.01

CLA: conjugated fatty acid; LA: linolenic acid; ALA: alpha linoleic acid; SE: standard error. Values with different superscripts within a row are significantly different at *p* ≤ 0.05.

**Table 7 animals-13-01317-t007:** Associations of haplotypes in different blocks with milk fat (%) and fatty acid profiles (g/100 g) in Najdi sheep.

Trait	HB	SNPs	Position	SE	r^2^	T	*p* Value
C16:1 cis7	H1B1	PIII- T > G	1007	0.009	0.061	2.16	0.03
	H4B1	PIII- G > T	1331	0.009	0.064	2.20	0.03
C22:0	H2B1	PIII- C > T	1014	0.031	0.089	2.59	0.01
C6:0	H1B2	PI- G > A	4412	0.079	0.052	2.00	0.04
	H2B2	PI- T > A	4441	0.078	0.053	2.03	0.04
C8:0	H1B2	PI- G > A	4412	0.079	0.052	2.00	0.04
	H2B2	PI- T > A	4441	0.078	0.053	2.03	0.04
ALA (n3)	H1B2	PI- G > A	4412	0.041	0.092	2.72	0.01
	H2B2	PI- T > A	4441	0.040	0.084	2.59	0.01
	H5B3	E53- C > T	6898	0.043	0.098	2.73	0.01
	H9B3	E53- C > T	7031	0.043	0.062	2.12	0.03
Fat %	H3B3	E53- G > C	6860	0.27	0.07	2.31	0.03
LA (n6)	H1B3	E53- G > C	6627	0.121	0.069	2.25	0.03
	H2B3	E53- C > T	6855	0.117	0.102	2.79	0.007
	H4B3	E53- T > C	6894	0.122	0.108	2.87	0.005
	H5B3	E53- C > T	6898	0.124	0.104	2.81	0.006
	H6B3	E53- C > G	6977	0.117	0.109	2.89	0.005
	H7B3	E53- T > C	6989	0.123	0.080	2.43	0.01
	H9B3	E53- C > T	7031	0.123	0.091	2.61	0.01

HB: haplotype block; PI: promoter I; PIII: promoter III; E53: Exon 53; SE: stander error; r^2^: pairwise SNP correlations; T: *t* test; *p* < 0.05.

## Data Availability

The data presented in this study are available on request from the corresponding author.

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
