# Peer review of "Identification of Genetic Polymorphisms of PI, PIII, and Exon 53 in the Acetyl-CoA Carboxylase-α (ACACα) Gene and Their Association with Milk Composition Traits of Najdi Sheep"

_animals, 2023, doi:10.3390/ani13081317_

Round 1

Reviewer 1 Report

The Najdi sheep are mainly found in the central and eastern regions of Saudi Arabia's. Under improved feeding and management practices, the Najdi breed performs satisfactorily and has high potential for milk production with intensive production. This study identifid SNPs in the ACACα gene PI, PIII, and exon 53, and their association with MC and FA profiles in Najdi sheep milk. 76 multiparous Najdi ewes were used.The identified association between Najdi breed genotypes and phenotypes could play a crucial role in altering milk composition and FA profile of milk fat, and could form the basis of a genetic selection program to produce healthy milk.

However, there are still some minor problems that need to be revised, as follows.

1 The number of individuals measured in this study is less, and the results of more group data will be more reliable, reflecting the size of the group in which the individuals are studied in the manuscript.

2 There are no descriptive statistics reflecting the distribution of milk components and fatty acids in the article. How many kinds of milk components and fatty acids have been detected in this study, and their situation is what readers want to know.

3 The pictures in this manuscript seem unclear, and the fonts in the pictures are not consistent.

4 Why should the same headers as Table 3 and Table 4, Table 5 and Table 6 be separated into two tables?

The design of this study is reasonable, the expression is clear and the results are reliable.

Author Response

Dear Prof,

Thank you for giving us the opportunity to submit a revised draft of our manuscript titled Identification of genetic polymorphisms of PI, PIII, and exon53 in the acetyl-CoA carboxylase-α (ACACα) gene and their association with milk composition traits of Najdi sheepto the Animals Journal. We appreciate the time and effort that you and the reviewers have dedicated to providing your valuable feedback on our manuscript. We are grateful to the reviewers for their insightful comments on our paper. We have been able to incorporate changes to reflect most of the suggestions provided by the reviewers. We have highlighted the changes within the manuscript. Based on this, we are optimistic that the revised version of the manuscript would achieve certain publishable status.

Reviewer 2 Report

Revision of manuscript entitled “Identification of genetic polymorphisms of PI, PIII, and exon53 in the acetyl-CoA carboxylase-α (ACACα) gene and their association with milk composition traits of Najdi sheep”. 

The manuscript deals with sequence analysis of different regions of the ACACα gene in a population of 76 Najdi sheep, and association analysis with milk composition traits.

The topic is interesting to the field.

The main weakness of the present manuscript is the number of animals, too low to make association analysis, and to draw conclusions.

The positive of the manuscript is the analysed population, which is local (and then interesting).

Some minor issues:

Line 9: “The acetyl-CoA carboxylase (ACACα) gene” please name correctly the gene before the short form (or correctly name the short form after the gene, in this case it is “ACC”).

Line 22-23_ please explain, the sheep were multiparous and you colleced milk during the first lactation, it does’nt make sense. “76 multiparous Najdi ewes were used and maintained on the same feeding system. Milk and blood samples were collected during the first lactation”.

Line 26: “vaccine acid (VA)” did you mean vaccenic acid?

Line 34: please provide at least one reference.

Line 45-49: at this point, you should state that mammals have at least an alpha and a beta ACAC genes, and if you are going to analyse the ACAC alpha gene in sheep, you should provide information on its location in the sheep genome (it would be useful to know how many exons it has).

Line 67: “and SNPs loci associated with MC and FA profiles” please rephrase.

Line 79-87: please describe the methods so that they can be repeated just using the present publication.

Line 102: please check this sentence “((Table 2). of a Final volume PCR)”

Line 207: “MT512519” indicating the accession number of the sequence submission is not sufficient to identify an SNP. It is necessary to use an identifier that you can find in the public databases (if it was already identified in other sheep) or you should submit it to a public database to obtain a unique identifier (EVA archive).

Line 211:” Hardy-Weinberg Frequencies (Expected)” what do you mean?

Line 214: “3.2. Association of genotype analyzes with milk traits” paragraph is difficult to read.

Tables are redundant in their content, please simplify (especially tables 5 and 6). Please provide a summary table describing milk traits.

Author Response

(The authors gave the same response as above.)

Round 2

Reviewer 2 Report

the manuscript is now much clearer, it deserves to be published on Animals